# Clinicopathological and Genomic Identification of Breast Cancers with No Impact on Mortality

**DOI:** 10.3390/cancers16061086

**Published:** 2024-03-07

**Authors:** Salvador Gámez-Casado, Lourdes Rodríguez-Pérez, Cristina Bandera-López, Andrés Mesas-Ruiz, Alicia Campini-Bermejo, Marta Bernal-Gómez, Manuel Zalabardo-Aguilar, Julio Calvete-Candenas, Gala Martínez-Bernal, Lidia Atienza-Cuevas, Marcial García-Rojo, Encarnación Benítez-Rodríguez, Bella Pajares-Hachero, María José Bermejo-Pérez, José M. Baena-Cañada

**Affiliations:** 1Medical Oncology Department, Hospital Universitario Puerta del Mar, 11009 Cádiz, Spain; salvagc85@hotmail.com (S.G.-C.); lourdesrodriguez1986@gmail.com (L.R.-P.); acampini@live.com (A.C.-B.); martabernal31@gmail.com (M.B.-G.); jjcalvete@gmail.com (J.C.-C.); galamartinezbernal@gmail.com (G.M.-B.); 2Instituto de Investigación e Innovación Biomédica de Cádiz (INIBICA), 11009 Cádiz, Spain; lidia.atienza.sspa@juntadeandalucia.es (L.A.-C.); marcial.garcia.sspa@juntadeandalucia.es (M.G.-R.); encarnacion.benitez@uca.es (E.B.-R.); 3Medical Oncology Department, Hospital Universitario Virgen de la Victoria, 29010 Malaga, Spain; cristina_bandera91@hotmail.com (C.B.-L.); mesasruiz91@gmail.com (A.M.-R.); m.zalabardo@uma.es (M.Z.-A.); bella.pajares@ibima.eu (B.P.-H.); cheberpe@gmail.com (M.J.B.-P.); 4Instituto de Investigación Biomédica de Málaga (IBIMA), 29010 Malaga, Spain; 5Pathology Unit, Hospital Universitario Puerta del Mar, 11009 Cádiz, Spain; 6Pathology Unit, Hospital de Jerez, 11407 Jerez, Spain; 7Preventive Medicine Department, Hospital Universitario Puerta del Mar, 11009 Cádiz, Spain

**Keywords:** low-risk breast cancer, metastasis, observational study, screening

## Abstract

**Simple Summary:**

There are patients with breast cancer which will never metastasize. Tumor palpability is a prognostic variable for lower risk of developing metastases. PAM50 intrinsic subtypes were independently prognostic for long-term survival. In our study, patients with non-palpable, luminal tumors, <1 cm, diagnosed on breast screening, never develop metastases. De-escalation of treatment should be considered.

**Abstract:**

Background. Implementing mammogram screening means that clinicians are seeing many breast cancers that will never develop metastases. The purpose of this study was to identify subgroups of breast cancer patients who did not present events related to long-term breast cancer mortality, taking into account diagnosis at breast screening, absence of palpability and axillary involvement, and genomic analysis with PAM50. Patients and Methods. To identify them, a retrospective observational study was carried out selecting patients without any palpable tumor and without axillary involvement, and a genomic analysis was performed with PAM50. Results. The probability of distant metastasis-free interval (DMFI) of 337 patients was 0.92 (95% CI, 0.90–0.93) at 20 years and 0.96 (95% CI, 0.92–1.00) in 95 patients (28%) with available PAM50 tests. In 22 (23.15%) luminal A tumors and in 9 (9.47%) luminal B tumors smaller than 1 cm, and in HER2 and basal type tumors, there were no metastatic events (20-year DMFI of 1.00). Conclusion. Patients with nonpalpable breast cancer found at screening with negative nodes are at very low risk. It is possible to identify subgroups without metastatic events by determining the intrinsic subtype and tumor size less than 1 cm. Therefore, de-escalation of treatment should be considered.

## 1. Introduction

The expansion in breast screening (mammogram) programs has resulted in clinicians seeing a large number of very early stage breast cancers [1]. Advertising emphasizes the benefits of detecting these preclinical lesions because intuition tells us that early stage treatment of breast cancer should facilitate treatment and cure [2].

Mammography, as an effective screening test for breast cancer, should not only increase the reported incidence of preclinical disease, but should also decrease the reported incidence of metastatic cancer and cancer-related mortality. Otherwise, early detection efforts may be uncovering a reservoir of nonprogressive, or very slowly progressive, breast cancers that were not destined to cause symptoms for the rest of the woman’s life, nor her death, a phenomenon known as overdiagnosis [3].

If cancer is detected, it is impossible to know who has been overdiagnosed, as it is not possible to know how it would have evolved in the absence of screening. However, from the point of view of the clinician caring for these patients with very early breast cancer detected on screening mammography, it would be essential to be able to identify those whose diagnosis will have no impact on mortality because this would allow their treatment to be de-escalated in some way. To do so, the first requirement would be to demonstrate that a particular patient, diagnosed with breast cancer, will not have metastases, or breast cancer-related death, for an extended period of time. Secondly, the clinical, pathological and genomic characteristics of their tumors should be analyzed and, thirdly, a clinical trial should be designed to de-escalate the treatments.

National registry studies can identify patients with breast cancer with no decrease in life expectancy [4], and patients with certain breast cancers may even have higher relative survival than the general population if the cancer is detected incidentally or by mammogram screening and patients also adopt other healthy behaviors (the healthy-user effect) [5].

Within the group of women diagnosed in breast screening, tumor palpability at diagnosis [6], a parameter that leads us to classify it as preclinical cancer, is a prognostic variable that helps to select those with a lower risk of developing metastases and with lower mortality [6]. Traditionally, breast cancer treatment decisions and prognosis have been based on tumor size and nodular extension. This is a subgroup of patients with very small tumors (less than 1 cm) and no axillary involvement, with a very good prognosis, and a very low risk of metastasis [7]. Another way to identify breast cancers without a clear impact on mortality is the genomic analysis of the breast tumor. In a large prospective study, patients with ultra-low genomic risk determined by 70-gene signature (Mammaprint) had the best prognosis, with a reduced risk of distant metastasis or breast cancer-related death [8]. PAM50 intrinsic subtypes were independently prognostic for long-term breast cancer survival [9].

The purpose of this study was to identify subgroups of breast cancer patients who did not present events related to long-term breast cancer mortality, taking into account diagnosis at breast screening, absence of palpability and axillary involvement, and genomic analysis with PAM50.

## 2. Materials and Methods

### 2.1. Study Design

A retrospective, observational study was carried out on breast cancer patients from a public or opportunistic mammogram screening program, with non-palpable breast tumors and without axillary involvement, treated in the Medical Oncology departments of two university hospitals between 2001 and 2014.

Data related to patients and treatment received were collected retrospectively from the clinical records of the Medical Oncology departments. The Spanish register of deaths (Índice Nacional de Defunciones) was consulted for vital status and patients’ date of death.

### 2.2. Inclusion and Exclusion Criteria

Women seen in the oncology clinic for breast carcinoma with the following characteristics: detected during the screening program at the asymptomatic phase with non-palpable tumor and with no other signs or symptoms directly related to the tumor, such as skin retraction, bloody nipple discharge, or others, and with negative axillary nodes in diagnostic surgery. Tumors with histologic grade 1, 2 and 3, estrogen and progesterone receptor positive and negative, human epidermal growth factor receptor 2 (HER2) negative and positive were permitted. Males and carcinomas in situ were excluded.

### 2.3. Immunohistochemistry and Molecular Studies

After 24 h of 10% formaldehyde fixation, breast tissues were routinely paraffin embedded and sliced into 3 μm continuous sections. For immunohistochemical staining, all samples were processed using a sensitive UltraView™ Universal DAB Detection Kit from Roche Diagnostics© (Basel, Switzerland) detection system in an automated Ventana Benchmark Ultra automatic stainer. The following rabbit monoclonal primary antibodies were used for staining, all of them from Roche Diagnostics© (Basel, Switzerland): CONFIRM anti-Ki-67 (clone 30-9), CONFIRM ER (clone SP1), CONFIRM PR (clone 1E2), and PATHWAY anti-HER-2/neu (clone 4B5).

Molecular analysis of gene expression of 50 genes (PAM50) allowed the tumor to be classified into one of 4 intrinsic subtypes: luminal A, luminal B, HER2-enriched and basal-like. Formalin-fixed, paraffin-embedded (FFPE) tumor tissue blocks and corresponding slides were obtained and reviewed by a pathologist who marked a representative area of the tumor. Tissue samples 1 mm in thickness were obtained from the area of the block corresponding to the marked slide. Two samples per box (or one sample if the primary tumor was less than 0.7 cm in diameter) were placed in plastic tubes with an identification number and sent to the reference laboratory. FFPE tissue samples were deparaffinized and digested for RNA extraction following a previously published procedure [9].

After assessment of their quality and concentration, the samples were processed and recorded in nCounter^®^. Samples were prepared with marker and capture probes, incubated in a thermocycler and transferred to the nCounter^®^ preparation station. The reading was performed using a digital analyzer. The analysis of the expression values was performed with the bioinformatics tool nSolver^®^.

### 2.4. Data Analysis

The events collected were ipsilateral local recurrence after conserving surgery, postmastectomy local recurrence, regional lymph node recurrence, distant metastases, second primary invasive tumors (in ipsilateral or contralateral breast and other locations) and ipsilateral or contralateral breast carcinoma in situ. The dependent variable analyzed was the distant metastasis-free interval (DMFI), defined as the interval between surgery and either distant metastasis or death from breast cancer [10] (events related to breast cancer mortality).

A descriptive analysis of the variables related to patients, breast tumor and treatment (absolute and relative frequency, mean, median and standard deviation) was performed. For the comparison of qualitative variables, the chi-square test with Fisher’s correction was applied and, for quantitative variables, the Student’s *t*-test was used.

The Kaplan–Meier method and the Log-Rank test for the comparison of survival curves were used to calculate survival. Cox regression with an independent variable was used to calculate the hazard ratio. SPSS version 21 was used for statistical analysis of the data. In the statistical analysis, *p* < 0.05 was considered to indicate statistical significance.

## 3. Results

### 3.1. Patient Characteristics

Three hundred and thirty-seven patients met the inclusion and exclusion criteria. In 31 cases (9.20%) the immunohistochemical subtype was not available, 268 (79.50%) cases were hormone-sensitive, 23 (6.80%) were triple-negative, and 15 (4.40%) were HER2. For 96 patients (28.40%) intrinsic subtyping was available with PAM50 (57, 60% luminal A; 26, 27% luminal B; 6, 6.30% HER2; 7, 7.40% basal-like) (Figure 1). In one luminal A subtype patient the information on events and survival was not available.

The median age was 56 years, with a range between 38 and 71. Participation in the public screening program was from the age of 45 up until 2013, and the presence of younger women in the study (65, 19.20% premenopausal) is explained as a result of opportunistic screening. All were referred from breast screening (302, 89.30% public and 36, 10.70% opportunistic). The vast majority had full functional capacity. Of the patients, 93.20% had no or few comorbidities. Most of the carcinomas (82.50%) were of the invasive ductal type, stage I (77.20%), with a median tumor size of 1.20 cm, mean histologic grade in half of the cases and a median Ki67 proliferative index of 12%. Estrogen receptors were positive in 88.50% of tumors and progesterone receptors in 76%. Regarding HER2, 4.40% were positive.

Some 88.80% of the women underwent breast-conserving surgery and 68.60% underwent axillary lymph node-sparing surgery. Among the patients, 73.10% were treated with adjuvant hormone therapy for 5 years, and one third with 3–6 months of adjuvant chemotherapy.

Table 1 shows the patients’ characteristics.

### 3.2. Event Analysis

The median follow-up of patients was 142 months (1–249). During this time, 72 patients suffered an event (21.30%), as shown in Table 2.

In the 22 patients with tumors of luminal intrinsic subtype A and less than 1 cm, there were no events related to breast cancer mortality. In the 34 patients with tumors of luminal intrinsic subtype A and greater than 1 cm, there were 2 events related to breast cancer mortality. Likewise, in the 9 patients with luminal intrinsic subtype B tumors and less than 1 cm, there were no events related to breast cancer mortality. Again, in the 17 patients with luminal B tumors and greater than 1 cm, there was 1 event related to breast cancer mortality.

In the 6 patients with intrinsic HER2 subtype and in the 7 with basal-like subtype, no metastatic events occurred (Table 3).

### 3.3. Survival Analysis

The distant metastasis-free interval probability at 5, 10, 15, and 20 years was 0.98 (95% CI, 0.96–0.99), 0.94 (95% CI, 0.91–0.97), 0.92 (95% CI, 0.90–0.93), and again 0.92 (95% CI, 0.90–0.93), respectively.

Figure 2 shows the distant metastasis-free probability curve.

In the 95 patients with available intrinsic subtype determination the distant metastasis-free interval probability at 5 years was 1.00 and at 10 years, 15 years, and 20 years was 0.96 (95% CI, 0.92–1.00).

Figure 3 shows the distant metastasis-free interval probability curve for the 95 patients with intrinsic subtype determined using PAM50.

## 4. Discussion

In this retrospective attempt to identify subgroups of breast cancer patients who did not present events related to long-term breast cancer mortality, taking into account diagnosis at breast screening, absence of palpable anomalies, and no axillary involvement, and genomic analysis with PAM50, we found that patients with luminal intrinsic subtypes A and B and tumors smaller than 1 cm (32.62%) never develop metastatic events. The low representation of HER2 (6.30%) and basal (7.30%) tumors also did not develop metastases. As has been shown before, patients with tumors smaller than 1 cm, with luminal A, HER2, and triple-negative subtypes on immunohistochemistry did not develop metastases [13].

The underlying biological basis for the 100% distant metastasis-free interval that are luminal tumors smaller than 1 cm, detected in mammogram screening, with non-palpable presentation and negative lymph nodes, would be nonprogressive preclinical cancers and we can also assume that, due to their biological characteristics, tumors with basal subtypes and HER2 would be progressive preclinical cancers and would become nonprogressive with treatment. Breast cancers detected on mammogram screening may be progressive or nonprogressive preclinical cancers [14]. The former will manifest as clinical cancers during the woman’s lifetime if no mammograms are carried out and could lead to death unless they are treated. Preclinical nonprogressive cancers would not present as clinical cancer during the woman’s lifetime if no mammograms were carried out and death would be as a result of a different cause. A woman with preclinical nonprogressive cancer is always overdiagnosed because the cancer never progresses to a clinical state and may even return [15].

Breast screening detects slow-growing cancers, the vast majority of which tend to be of the luminal subtype. In our study, the proportion of HER2 or basal-like breast cancers has been lower than the known proportion for all age groups [16,17]. Specifically, the age, represented by a low number of young women in our study, where the presentation of these aggressive tumor subtypes is more frequent, justifies this lower proportion of HER2 and basal tumors. Likewise, the aggressive behavior of these rapidly growing tumor subtypes with a shorter preclinical detectable phase justifies their lower frequency in our study, as these tumors often emerge clinically between screening cycles [17]. HER2 and basal cancers detected using screening mammography are usually diagnosed at an earlier stage, and their prognosis was better than those detected by symptoms [18].

As we said in the introduction, the next step, once the subgroups of breast cancer patients who, having undergone standard treatment, will not develop metastases have been identified, would be to de-escalate the treatment.

For over two decades, the PROTECT clinical trial [19] has been assessing the efficacy of treatments compared to active monitoring among men with clinically localized prostate cancer detected using prostate specific antigen (PSA) screening. A 15-year analysis provides evidence of a high long-term survival rate in the trial population (3% prostate cancer-specific mortality and 22% mortality from another cause), regardless of treatment group. Although radical treatments (prostatectomy or radiotherapy) have halved the incidence of metastases, local progression, and androgen deprivation therapy compared with active monitoring, these reductions did not translate into differences at the 15-year mortality rate [19]. This hypothesis should reasonably translate to the breast cancer setting and should be confirmed with randomized prospective studies that include some form of de-escalation of treatment.

If we exclude progressive preclinical cancers [20] (HER2 and basal), the adjuvant systemic treatment of women with luminal tumors and tumors smaller than 1 cm is hormone therapy. De-escalating this treatment would be possible in this subgroup of patients because the effect of hormone therapy on the development of metastases is limited [21]. In fact, in the Mindact trial, which had an 8-year follow-up, the distant metastasis-free interval and overall survival of low-risk stage I patients treated with adjuvant hormone therapy was not statistically different from those who did not receive this treatment [22], using propensity score matching methodology to select a group of patients receiving hormone therapy and another not receiving hormone therapy. A drawback of refraining from using adjuvant hormonal therapy is that this treatment not only reduces distant relapses, but also locoregional recurrences and second contralateral or ipsilateral breast cancers [21]. As we have seen in the event analysis of our study, the number of locoregional recurrences and second primary breast tumors is similar to the number of metastases. The follow-up period of the Mindact trial of 8 years and that of our study of almost 12 years seem short due to the considerable risk of late recurrence of these luminal tumors, with recurrences up to 32 years after diagnosis [22]. For this reason, it is impractical to design a controlled clinical trial to answer research questions that require decades of follow-up and also require events that will almost never occur.

In fact, at present, de-escalation measures could already be applied in this patient population. For example, based on the SOUND non-inferiority trial [23], the results of which suggest that breast cancer patients with tumors of less than 2 cm and ultrasonographically negative axillary lymph nodes, can safely avoid any axillary surgery [23], the patients identified in our study could also be offered the choice to abstain from axillary surgery.

Likewise, women with luminal A tumors could also be offered abstinence from adjuvant radiotherapy after conserving surgery, based on the results of the LUMINA trial [24].

Finally, as we have seen [21], in patients with low-risk stage I breast cancer, the beneficial effect of hormone therapy on the distant metastasis-free interval is limited and has to be counterbalanced by its side effects [25], and should be discussed with these patients at very low risk of distant metastasis [21,25]. The Canadian LA LEAST trial compares 2 years of endocrine therapy in women over 50 years of age whose node-negative breast cancers are low risk according to the Prosigna score [26], also as a form of de-escalation. However, one must be very careful about combining several forms of de-escalation at the same time, for example, omitting radiotherapy and hormone therapy at the same time [27]. Just like for ductal carcinoma in situ [28], dose reduction of hormone therapy could be explored in this group of very low-risk patients. There are no data with dose reduction of aromatase inhibitors, but there are data with reduction to 5 mg per day of tamoxifen [28].

Lastly, the risk of death from breast cancer is also related to the patient’s age [29], so the de-escalation measures discussed may not be applicable to all age groups.

The strengths of this study include its large cohort of patients with long-term follow-up and reliable data based on individual medical records. However, this study suffered from biases, such as the low availability of material for the determination of the PAM50 genomic study and the biases associated with retrospective studies, such as selection bias. It is possible that the period when patients were diagnosed, without the use of new technology such as digital mammography and tomosynthesis, may influence the number of preclinical cancers detected [30]. The results from a more up-to-date study could have been different.

## 5. Conclusions

Our study provides novel insights into the epidemiology of very low-risk breast cancer. Oncologists and other health care professionals should be aware that there are patients with breast cancer who will never metastasize, particularly women diagnosed on breast screening with non-palpable, luminal subtype tumors smaller than 1 cm and node negative. Our study advocates further research to optimize treatment de-escalation to prevent overtreatment.

## Figures and Tables

**Figure 1 cancers-16-01086-f001:**
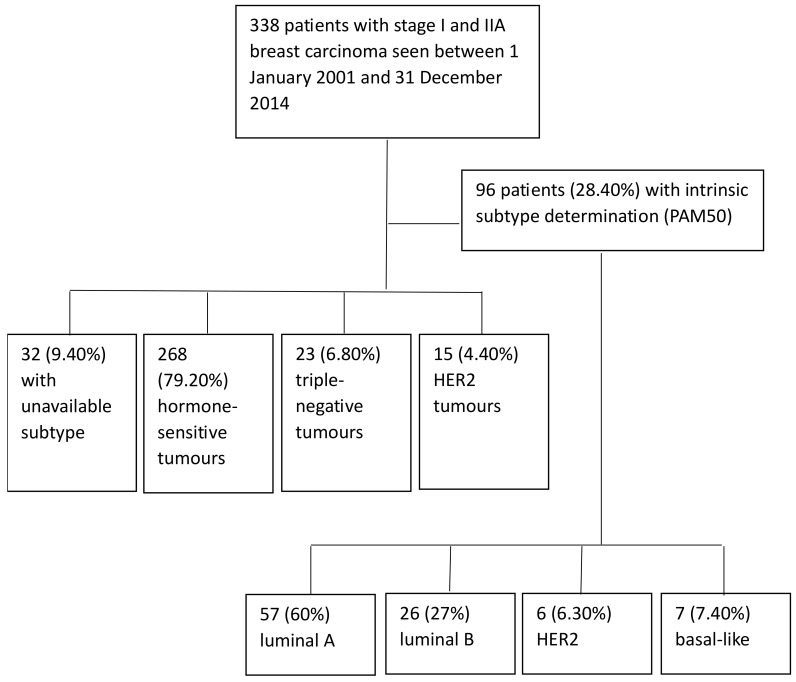
Flow chart of the study.

**Figure 2 cancers-16-01086-f002:**
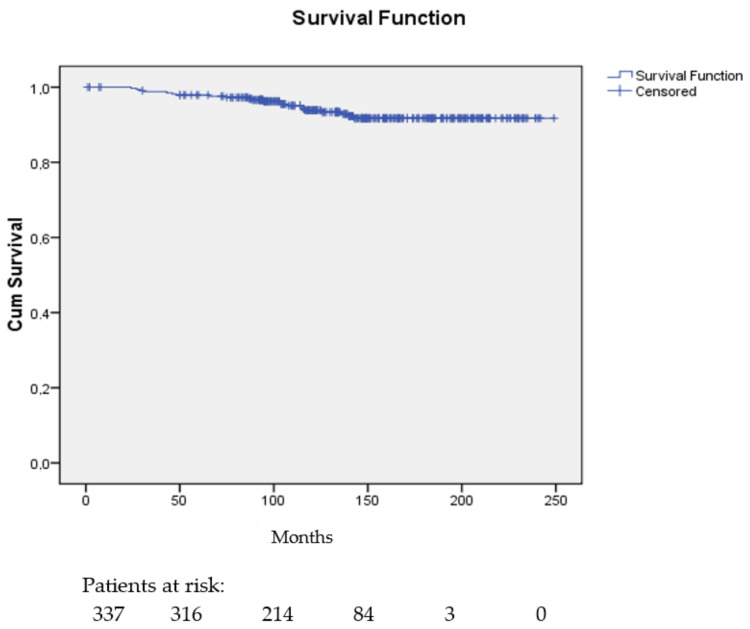
Distant metastasis-free interval probability curve for the 337 patients.

**Figure 3 cancers-16-01086-f003:**
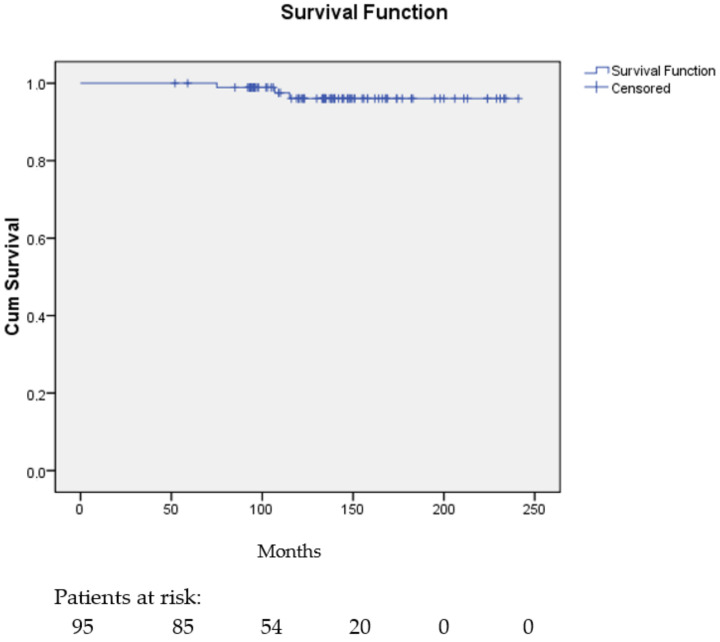
Distant metastasis-free interval probability curve for 95 patients with intrinsic subgroup determination by PAM50.

**Table 1 cancers-16-01086-t001:** Characteristics of the 338 patients in the study.

Characteristics of the 338 Patients	N	%
Age, median (range)	56 (38–71)	
Functional capacity (ECOG) ^1^		
0	296	87.60
1	40	11.80
2	1	0.30
3	1	0.30
Comorbidity ^2^		
0	83	24.60
1	21	6.20
2	211	62.40
3	20	5.90
4	1	0.30
6	2	0.60
Menopause status		
Premenopausal	65	19.20
Postmenopausal	273	80.80
Breast screening mammogram		
Public	302	89.30
Opportunistic	36	10.70
Histological type		
Ductal	279	82.50
Lobular	21	6.20
Others	38	11.20
Stage		
I	261	77.20
IIA	75	22.20
Unknown	2	0.60
Tumor (pT)		
pT1mi	8	2.40
pT1a	21	6.20
pT1b	101	29.90
pT1c	161	47.60
pT2	44	13
Unknown	3	0.90
Tumor size (cm), median (range)	1.20 (0.06–5)	
Histological grade		
1	106	31.40
2	167	49.40
3	49	14.50
Unknown	16	4.70
Ki67 proliferative index, median (range)	12 (1–90)	
Estrogen receptors		
Positive	299	88.50
Negative	34	10.10
Unknown	5	1.50
Progesterone receptors		
Positive	258	76.30
Negative	75	22.20
Unknown	5	1.50
HER2		
Positive	15	4.40
Negative	318	94.10
Unknown	5	1.50
Breast surgery		
Conserving	300	88.80
Mastectomy	38	11.20
Axillary surgery		
Sentinel lymph node biopsy	232	68.60
Axillary lymphadenectomy	98	29
None	8	2.40
Adjuvant systemic treatment		
Hormone therapy	152	45
Chemotherapy	28	8.30
Chemotherapy-hormone therapy	95	28.10
None	58	17.20
Unknown	5	1.50
Hormone therapy		
Tamoxifen	106	43.60
Aromatase inhibitor	59	24.30
Tamoxifen-aromatase inhibitor	76	31.30
Ovarian ablation	2	0.80
Chemotherapy		
Anthracyclines	67	54
Anthracyclines and taxanes	37	29.80
Others	20	16.10

^1^ Measured by the ECOG (Eastern Collaborative Oncology Group) scale [11]; ^2^ Measured by the Charlson scale [12].

**Table 2 cancers-16-01086-t002:** Events in the 338 patients.

Events	N	%
Local recurrence	16	4.70
After conserving surgery	13	3.80
After mastectomy	3	0.90
Regional recurrence	3	0.90
Metastasis	18	5.30
Liver	6	1.80
Bone	5	1.50
Lung	5	1.50
Central nervous system	2	0.60
Skin	1	0.30
Peritoneum	1	0.30
Others	2	0.60
Secondary invasive primaries	31	9.30
Breast	10	3
Ipsilateral	3	0.90
Contralateral	7	2.10
Non-breast	21	6.30
Colorectal	4	1.20
Ovarian	3	0.90
Endometrial	3	0.90
Pancreatic	2	0.60
Lung	1	0.30
Thyroid	1	0.30
Bladder	1	0.30
Vulva	1	0.30
Carcinoid tumor	1	0.30
Soft tissue sarcoma	1	0.30
Non-Hodgkin’s lymphoma	1	0.30
Multiple myeloma	1	0.30
Oncocytoma	1	0.30
Secondary non-invasive primaries	7	2.10
Ductal carcinoma in situ of the ipsilateral breast	5	1.50
Ductal carcinoma in situ of the contralateral breast	2	0.60
Deaths	30	9
From breast carcinoma	16	4.80
From other causes	12	3.60
Due to unknown causes	2	0.60

**Table 3 cancers-16-01086-t003:** Metastatic and non-metastatic events in relation to intrinsic subtype and tumor size less than or greater than 1 cm.

Intrinsic Subtype and Tumor Size	N and %	Events (N and %)	Events Not Related to Breast Cancer Mortality	Events Related to Breast Cancer Mortality
Luminal A	56/95 (58.90%)	9/56 (16.10%)	7/56 (12.50%)	2/56 (3.50%)
<1 cm	22 (39.30%)	3/56 (5.30%)	3/56 (5.30%)	0/56 (0%)
>1 cm	34 (60.70%)	6/56 (10.70%)	4/56 (7.10%)	2/56 (3.60%)
Luminal B	26/95 (27.30%)	6/26 (23%)	5/26 (19.20%)	1/26 (3.80%)
<1 cm	9 (34.60%)	3/26 (11.50%)	3/26 (11.50%)	0/26 (0%)
>1 cm	17 (65.40%)	3/26 (11.50%)	2/26 (7.70%)	1/26 (3.80%)
HER2 Enrichment	6/95 (6.30%)	1/6 (16.70%)	1/6 (16.70%)	0/6 (0%)
<1 cm	0 (0%)			
>1 cm	6 (100%)	1/6 (16.70%)	1/6 (16.70%)	0/6 (0%)
Basal like	7/95 (7.30%)	1/7 (14.30%)	1/7 (14.30%)	0/7 (0%)
<1 cm	4 (57.10%)	0/7 (0%)	0/7 (0%)	0/7 (0%)
>1 cm	3 (42.90%)	1/7 (14.30%)	1/7 (14.30%)	0/7 (0%)

## Data Availability

The data presented in this study are available in this article.

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
