# Peer review of "Clinicopathological and Genomic Identification of Breast Cancers with No Impact on Mortality"

_cancers, 2024, doi:10.3390/cancers16061086_

Round 1
Reviewer 1 Report
Comments and Suggestions for Authors
Gamez-Casado et al describe a retrospective analysis of breast cancer patients that presented without palpable tumors and axillary involvement and conducted a PAM50 analysis on some of these tumors to further identify their PAM50 subtypes. Distant metastasis-free survival interval within this cohort of patients for each subtype was calculated and reported. The observation and the results of the study are intuitive, but not novel. The authors report on the metric of metastasis but fail to hypothesize any underlying biological rationale that would be able to predict distant metastasis-free survival of new patients. These datasets are rich sources of information, and thus can be used to test various hypotheses of what factors may or may not contribute to metastasis. Additionally, as for materials and methods, the authors must provide the protocol for how the PAM50 analysis was conducted as well as the immunohistochemistry. The authors do not outline what antibodies or reagents were used in their assays.
Reviewer 2 Report
Comments and Suggestions for Authors
Gámez-Casado et al reported the results of their retrospective studies of breast cancer patients who could have been over-diagnosed and possibly over-treated. They proposed a solution and criteria to consider de-escalation of treatment for these patients. The topic is important; however, the studies may have some limitations and the manuscript needs to improve.
General comments:
1. The film-screen technology, outdated, many obtained only single views of each breast and there were limited screening rounds with variable screening intervals. Today, digital mammography with tomosynthesis, which acquires dozens of views of each breast, has increased cancer detection, and a decrease in callbacks and false positives is standard. The retrospective data sets might have used the outdated technology that does not reflect the nowadays state of the art technology.
2. The risk of diagnosis and dying from breast cancer is highly related to patients’ age. The study data set has limited numbers of patients; thus, the proposed solution may not apply to all patients of all age groups.
Specific comments:
1. In Figure 1. It is mentioned that there were a total of 338 patients with stage I and IIA breast carcinoma, with 96 patients having their intrinsic subtype determined. Figure 2 indicates 337 patients at risk, while Figure 3 displays 95 patients with intrinsic subtype determined. It would be helpful to clearly state how and why specific patients were excluded from the analysis.
2. The format of Figures 2 and 3 appears to be disordered.
3. In Figures 2 and 3, where the data range between 0.8 and 1.0, it might be more straightforward to use the range 0.8 to 1.0 instead of 0.0 to 1.0 on the vertical axis.
Reviewer 3 Report
Comments and Suggestions for Authors
13 February 2024
Ms. Ref. No.: cancers-2880706
Journal: Cancers
Title: Clinicopathological and genomic identification of breast cancers with no impact on mortality
Comments:
Thank you for taking the time to write this article on such a relevant topic. I found it to be informative and with the potential for further research in the future. However, I have some observations that I believe could be useful for improving the article, which I have mentioned in the following paragraphs.
1- In regard to article, patients with non-palpable, luminal tumours, < 1 cm, diagnosed on breast screening, never develop metastases. What is the reference of this criteria that luminal tumours, < 1 cm ?
2- The base of this research is retrospective observational study, if the size of tumors of participants will be increased after that, how can interpret the results?
3- How much the role of age range of patients could be influence the conclusion?
4- How was determine the sample size of this study?
5- Please reintroduce the association of this article with both of gastric and prostate cancer.
It is possible to enhance the introduction section's readability by including the following reference:
· https://doi.org/10.3390/cancers16040742
· https://doi.org/10.1007/s12013-023-01171-y
· https://doi.org/10.3390/cancers16040738
Round 2
Reviewer 2 Report
Comments and Suggestions for Authors
The authors satisfying addressed my comments.